# Does Motor Tract Integrity at 1 Month Predict Gait and Balance Outcomes at 6 Months in Stroke Patients?

**DOI:** 10.3390/brainsci11070867

**Published:** 2021-06-29

**Authors:** SoYeon Jun, BoYoung Hong, YoungKook Kim, SeongHoon Lim

**Affiliations:** 1Department of Rehabilitation Medicine, Seoul St. Mary’s Hospital, College of Medicine, The Catholic University of Korea, Seoul 06591, Korea; iamsj17@naver.com; 2Department of Rehabilitation Medicine, St. Vincent’s Hospital, College of Medicine, The Catholic University of Korea, Seoul 06591, Korea; byhong@catholic.ac.kr; 3Department of Rehabilitation Medicine, Yeouido St. Mary’s Hospital, College of Medicine, The Catholic University of Korea, Seoul 06591, Korea; england2@catholic.ac.kr

**Keywords:** stroke, gait, balance, diffusion tensor imaging, corticospinal tract, corticoreticular pathway, corticopontocerebellar tract

## Abstract

Recovery of balance and gait ability is important in stroke patients. Several studies have examined the role of white matter tracts in the recovery of gait and balance, but the results have been inconclusive. Therefore, we examined whether the integrity of the corticospinal tract (CST), corticoreticular pathway (CRP), and cortico-ponto-cerebellar tract (CPCT) at 1 month predicted balance and gait function 6 months after stroke onset. This retrospective longitudinal observational clinical study assessed 27 patients with first-ever unilateral supratentorial stroke. The subjects underwent diffusion tensor imaging 1 month after the stroke, and the Functional Ambulation Categories (FAC) and Berg Balance Scale (BBS) scores were assessed after 6 months. The normalized fiber number (FN) and fractional anisotropy (FA) results for the CST, CRP and CPCT were also obtained. The FN and FA results for the CST, CRP, or CPCT at 1 month were not related to the gait or balance at 6 months. There was also no difference in FAC values at 1 month after stoke onset among three groups differing in degree of independence of ambulation. The integrity of the CST, CRP, and CPCT on 1 month after stroke onset was not associated with gait or balance after 6 months. The white matter integrity did not predict the clinical outcome.

## 1. Introduction

Restoration of gait and balance is important for stroke patients. Well control of gait and posture is complex, and is managed by higher centers in the brain (locomotor programming at the level of the cerebral cortex, with basal ganglia and cerebellum involvement) [1]. Recovery of gait has been investigated in several ways. The studies with brain imaging revealed that lesion size and involvement of the internal capsule, caudate nucleus, or putamen white matter were associated with poorer gait recovery [2,3,4]. Other research showed that white matter injury might play roles in balance and gait function [5,6].

The corticospinal tract (CST) is involved in the recovery of upper limb motor function, especially hand function [7,8]. The corticoreticular pathway (CRP) is an extrapyramidal motor pathway that innervates the proximal muscles of the limbs and the axial muscles; it plays roles in standing, gait, and postural control [5,9]. The cerebellum is responsible for motor coordination and gait control, while the cerebellar afferent cortico-ponto-cerebellar tract (CPCT) delivers information on voluntary motor activity to the dentate nucleus [5,10]. CPCT injury was observed in a patient with ataxia and tremor after a traumatic brain injury [11].

We hypothesized that the integrity of these three white matter tracts is related to the recovery of gait and balance in patients with stroke, and investigated whether the integrity of the CST, CRP, and CPCT at 1 month after stroke onset predicts gait and balance outcomes at 6 months.

## 2. Materials and Methods

### 2.1. Study Design and Participants

This retrospective, longitudinal and observational clinical trial enrolled 27 patients suffering unilateral supratentorial first-ever stroke between September 2014 and September 2019. All patients were older than 18 years, had undergone 3T magnetic resonance imaging (MRI) with diffusion tensor imaging (DTI) 1–6 weeks (i.e., ~1 month) after stroke onset, and underwent functional assessments after 6 months. Exclusion criteria included underlying degenerative brain disease or other psychiatric disorder, an inability to ambulate before the stroke, and stroke recurrence within 6 months after onset.

Demographic, clinical, and brain DTI data were obtained from all patients. The 3T-MRI DTI was performed 24.9 ± 8.03 days after stroke onset. A rehabilitation program including neurodevelopmental physical and occupational therapy was treated within 7 days of stroke onset in all subjects and continued for up to 6 months (2–3 h daily, 5 days a week, with speech therapy as needed) [12].

Since this was an observational study, the number of samples was not specified in advance. However, since previous DTI parameter-based studies enrolled 10 to 25 subjects [13,14], we wanted to achieve a sample size of more than 25 subjects. Initially, 140 patients were seen during the study period, but 66 were excluded due to missing 1-month MRI data, while 21 had underlying lesions warranting exclusion, and 26 were lost to follow-up within 6 months after stroke onset. 

The study protocol was approved by the Institutional Review Board of Catholic University, College of Medicine (Registry No. VC19RISI0265), which waived the requirement for informed consent.

### 2.2. Functional Assessment

The demographic data of all subjects were reviewed retrospectively. The patients were functionally assessed 6 months after the stroke.

We classified the 27 patients into three groups according to their gait and balance. Gait was assessed using the Functional Ambulation Categories (FAC) scale, on which scores range from 0 to 5 according to the degree of independence in ambulation [15]. The patients were grouped by FAC score as follows: FAC score of 0–1, nonfunctional; FAC score of 2–3, dependent; and FAC score of 4–5, independent. patients were also grouped by Berg Balance Scale (BBS) score (0–20; very high, 21–40; moderate, or 41–56; low risk of falling) [15,16]. The Korean version of the modified Barthel Index (K-MBI) was used to estimate disability in all subjects [17,18].

### 2.3. Fiber Bundle Tracking

DTI was performed using a 3.0-T MAGNETOM^®^ Verio scanner (Siemens, Erlangen, Germany) equipped with a six-channel head coil. Data were acquired as single-shot spin-echo echo-planar images, with axial slices covering the entire brain across 76 interleaved 2.0-mm-thick slices [no gap; repetition time/echo time = 14,300/84 ms; field of view = 224 × 224 mm^2^; matrix = 224 × 224; voxel size = 1 × 1 × 2 mm^3^ (isotropic); number of excitations = 1]. Diffusion sensitizing gradients were applied in 64 noncollinear directions with a b-value of 1000 ms/mm^2^. The b = 0 images were scanned before acquiring the diffusion-weighted images, and there were 65 volumes in total [19]. DTI Studio (Johns Hopkins University, Baltimore, MD, USA) was used to compute the diffusion tensor for fiber bundle tracking, which was based on the fiber assignment by continuous tracking (FACT) algorithm with a fractional anisotropy (FA) threshold of 0.25 and angle threshold of 60 degrees [20]. 

Figure 1, Figure 2 and Figure 3 show the regions of interest (ROIs) used to reconstruct the CST, CRP, and CPCT in two-dimensional color maps. For reconstruction of CST there were seed and target ROI on the CST portion of the pontomedullary junction and the mid-pons, respectively (Figure 1) [10,21]. To reconstruct CRP, a seed ROI was placed on the CRP portion of the reticular formation in the medulla. On the midbrain tegmentum and the premotor cortex, there were first and second target ROI, respectively (Figure 2) [22]. For the CPCT, the portions of the CPCT on the middle cerebellar peduncle and the cerebral peduncle of the contra side were used as the seed and target ROI, respectively (Figure 3) [10] The ROIs mentioned above were used in a 2D color map. Fiber number (FN) and FA results of the CST, CRP, and CPCT were obtained in both the affected and unaffected hemispheres in all subjects. In the CPCT and non-CPCT groups, FN and FA were normalized by dividing the data for the affected side by that for the unaffected side [23].

### 2.4. Statistical Analysis

FN and FA are presented as the median (interquartile range). Fisher’s exact test was used to compare the demographics among the groups. When neither the normality of distribution nor equality of variance assumptions were satisfied, the Kruskal–Wallis test followed by the Mann–Whitney U test (with the Bonferroni correction) was used to test for differences among the groups in BBS scores and FAC. For all results, two-tailed *p*-values ≤ 0.0166 were deemed significant. The Wilcoxon signed-rank test was used to analyze the FAC on 1 and 6 months after stroke onset and to confirm the differences of FN and FA values for the three tracts (i.e., CST, CRP, and CPCT) between affected and unaffected lesion side on 1 month after the onset in all subjects. Since the variables were nonparametric groups and contain an ordinary scale, the correlation of functional assessment scores and DTI parameter (i.e., FN and FA) was assessed using Kendall’s tau b correlation. Additionally, the associations between the DTI parameters on 1 month after stroke onset and functional assessment on 6 month after stroke onset were determined through logistic regression analyses. All statistical analyses were performed using SPSS for Windows (ver. 21.0; SPSS, Chicago, IL, USA).

## 3. Results

Table 1 summarizes the demographic and clinical characteristics of the three groups. There were no significant group differences in age, gender, stroke type, lesion location, or affected hemisphere. Figure 1, Figure 2 and Figure 3 show representative results of diffusion tensor tractography of the CST, CRP, and CPCT. There were significant differences of FN and FA between affected and unaffected side CST, CRP and CPCT (all *p*-values < 0.001). Also FAC on 6 months showed marked improvement compared to that on 1 month. Table 2, Table 3 and Table 4 shows the FN and FA data for the CST, CRP, and CPCT by groups based on BBS and FAC scores. There were no significant associations of FN and FA with balance ability or gait on 1 and 6 months after onset. Only the BBS on 6 month after onset and FA value of CST showed a low level of correlation, and no other significant correlation of FN and FA with balance or gait ability 1 and 6 months after onset (Table 5). In logistic regression that included age and DTI parameters as covariates, there was no significant factor other than age for the balance or gait ability on 6 month after the stroke onset.

## 4. Discussion

The CSR, CRP, or CPCT at 1 month were not related to the gait and balance outcomes of patients with stroke at 6 months. The processing of somesthetic graviceptive sensation plays a critical role in the control of body orientation with respect to gravity, and thus in gait and balance, along with the motor tracts [24]. Reports suggest that injury to the insular cortex, postcentral gyrus, and posterior thalamus is associated with poor vertical posture [25]. A recent study showed that a larger stroke, rather than specific brain lesions, was more important with respect to the likelihood of restoration of vertical posture, where multiple sensory and motor systems are involved in the restoration of balance in stroke patients [26]. Balance function is a whole-brain phenomenon, rather than being associated with any specific part of the brain [27]. Previous studies, and our results, indicated that the CST, CRP, and CPCT are not associated with the recovery of gait and balance, just as no specific lesions affected the recovery of vertical posture [26]. The neuroradiological findings in the white matter alone did not predict the clinical outcomes in this study. These findings differ from those of some previous studies. There was a suggestion that CST independently would predict the response to gait rehabilitation in a study using the voxel based lesion symptom mapping (VLSM) [28]. Others presented that ipsilateral CST, CRP, and contra-lateral superior cerebellar peduncle(SCP) as biomarkers for working recovery [5]. However, our study is somewhat differentiated from those previous studies in that we targeted patients with various gait functions at the time of enrollment and analyzed both FN and FA values as DTI parameters.

Because ours was a small study, several statistical methods were used to overcome bias. We analyzed the subjects longitudinally, i.e., for up to 6 months after stroke onset [29]. We also examined three motor tracts to rule out over-estimation of any one tract analyzed. However, there were some limitations in our study including the small sample size and retrospective nature of the study. Also, this study included the inherent limitations from the functional assessment and diffusion tensor tractography used in this study. There was significant floor and ceiling effects on BBS despite of its useful properties and it has limited discriminatory power [16]. Acquisition disturbances or technical restrictions in the process of tractography should still be considered. A large-scale prospective longitudinal study using diverse functional assessments and other image tools is needed to address remaining questions and determine the reasons for the differences in results between previous studies and our investigation.

## 5. Conclusions

The integrity of the CST, CRP, and CPCT, as assessed by DTI about 1 month after a supratentorial first-ever stroke, did not reflect the functional outcome of gait and balance after 6 months.

## Figures and Tables

**Figure 1 brainsci-11-00867-f001:**
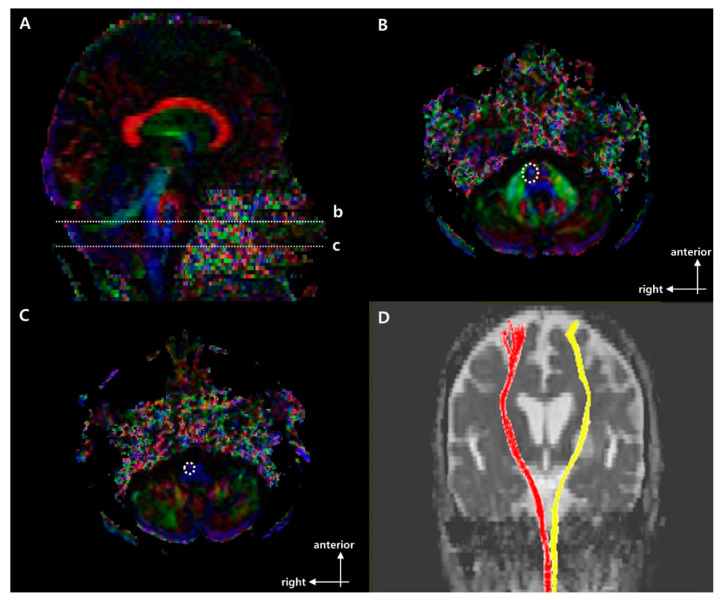
Regions of interest (ROIs used to reconstruct the right corticospinal tract (CST) on DTI and representative diffusion tensor tractography (DTT) images of corticospinal tract (CST). (**A**) Sagittal color fractional anisotropy (FA) map with dotted lines at (b) the mid-pons and (c) the pontomedullary junction. (**B**) The dotted line indicates the target ROI of the CST at the mid-pons in the axial color map (level b in (**A**)). (**C**) The dotted line indicates the seed ROI of the CST at the pontomedullary junction in the axial color map (level c in (**A**)). (**D**) Representative DTT image of CST in a typical subject with a left middle cerebral artery (MCA) infarction (red, unaffected side; yellow, affected side).

**Figure 2 brainsci-11-00867-f002:**
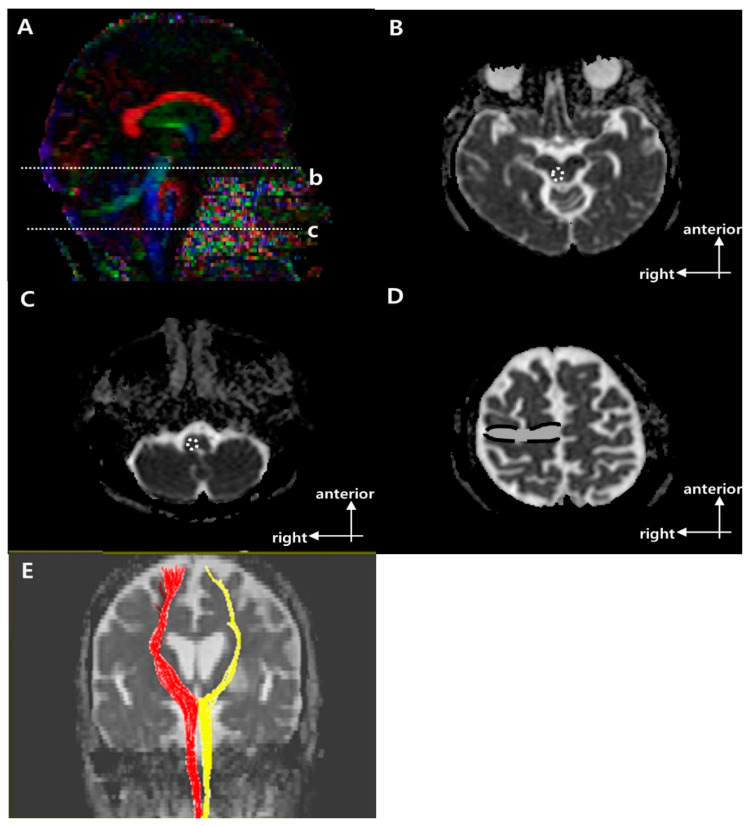
Regions of interest (ROIs) used to reconstruct the right corticoreticular pathway (CRP) on diffusion tensor imaging (DTI) and representative diffusion tensor tractography (DTT) images of the CRP [9]. (**A**) Sagittal color fractional anisotropy (FA) map with dotted lines showing the (b) reticular formation of the medulla and (c) midbrain tegmentum. (**B**) The dotted circle indicates the target ROI of the CRP at the midbrain tegmentum in the axial b = 0 image (level b in (**A**)). (**C**) The dotted circle indicates the seed ROI of the CRP at the reticular formation of the medulla in the axial b = 0 image (level c in A). (**D**) The dotted line indicates the second target ROI of the CRP in Brodmann area 6 on axial b = 0 image. (**E**) Representative DTT image of CRP in a typical subject with a left middle cerebral artery (MCA) infarction (red, unaffected side; yellow, affected side).

**Figure 3 brainsci-11-00867-f003:**
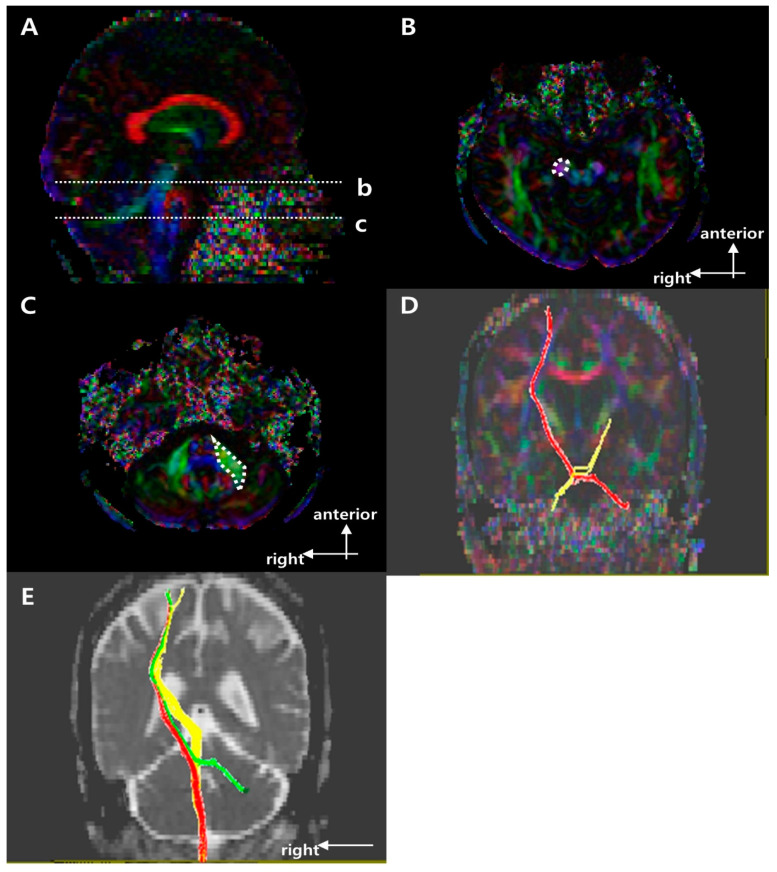
Regions of interest (ROIs) used to reconstruct the right cortico-ponto-cerebellar tract (CPCT) on diffusion tensor imaging (DTI) and representative diffusion tensor tractography (DTT) images of the CPCT. (**A**) Sagittal color fractional anisotropy (FA) map with the dotted lines showing the (b) cerebral peduncle and (c) mid pons. (**B**) The dotted circle indicates the target ROI of the CPCT at the cerebral peduncle in the axial color map (level b in (**A**)). (**C**) The dotted polygon indicates the seed ROI of the CPCT at the middle cerebellar peduncle at the mid pons in the axial color map (level c in (**A**)). (**D**) Representative DTT image of the CPCT in a typical subject with a left middle cerebral artery (MCA) infarction (red, unaffected side; yellow, affected side). (**E**) Representative superposing DTT image of the right side motor tracts in a typical subject with a left MCA infarction (red, CST; yellow, CRP; green, CPCT) in b = 0 image.

**Table 1 brainsci-11-00867-t001:** Participants’ demographic data.

Fall Risk→	High (0 ≤ BBS ≤ 20)	Moderate (21 ≤ BBS ≤ 40)	Low (41 ≤ BBS ≤ 56)	*p*-Value
Numbers of subjects	6	8	13	
Age, years	61.0 (56.3–66.5)	75.5 (62.5–81.0)	63.5 (33.5–73.8)	0.060
Gender				0.090
Female, *n* (%)	1 (16.7)	6 (75.0)	4 (30.8)	
Male, *n* (%)	5 (83.3)	2 (25.0)	9 (69.2)	
Stroke type				0.206
Infarction, *n* (%)	5 (83.3)	8 (100.0)	9 (69.2)	
Hemorrhage, *n* (%)	1 (16.7)	0 (0.0)	4 (30.8)	
Brain injury location				0.778
Cortex, *n* (%)	0 (0.0)	0 (0.0)	1 (7.7)	
Subcortex, *n* (%)	6 (100.0)	7 (87.5)	12 (92.3)	
Mixed, *n* (%)	0 (0.0)	1 (12.5)	0 (0.0)	
Hemispheric lesion				0.889
Left, *n* (%)	4 (66.7)	4 (50.0)	7 (53.8)	
Right, *n* (%)	2 (33.3)	4 (50.0)	6 (46.2)	
K-MBI	23.0 (12.0–34.8)	63.5 (33.5–73.8)	81.0 (77.0–84.0)	

Values are the median (interquartile range) or *n* (%). *p*-values were calculated using Fisher’s exact test or the Kruskal–Wallis test, followed by the Mann–Whitney *U* test with the Bonferroni correction. BBS, Berg Balance Scale; K-MBI, Korean version of the modified Barthel Index.

**Table 2 brainsci-11-00867-t002:** FN and FA values for the CST, CRP, and CPCT by groups classified based on the BBS score 6 months after stroke onset.

Fall Risk→		High (0 ≤ BBS ≤ 20, *n* = 6)	Moderate (21 ≤ BBS ≤ 40, *n* = 8)	Low (41 ≤ BBS ≤ 56, *n* = 13)	*p*-Value
CST	FN	0.043 (0.003–0.495)	0.414 (0.362–0.677)	0.392 (0.124–0.546)	0.214
	FA	0.644 (0.134–0.898)	0.882 (0.846–0.964)	0.948 (0.862–0.971)	0.058
CRP	FN	0.379 (0.275–0.480)	0.280 (0.193–0.636)	0.354 (0.211–0.483)	0.920
	FA	0.992 (0.935–1.033)	0.959 (0.940–0.978)	0.962 (0.934–0.977)	0.506
CPCT	FN	0.123 (0.018–0.443)	0.052 (0.0–0.465)	0.162 (0.0–0.351)	0.960
	FA	0.927 (0.230–0.983)	0.476 (0.0–1.019)	0.915 (0.0–0.965)	0.878

Values are the median (interquartile range). FN and FA were calculated as (affected side value/unaffected side value). *p*-values were tested using the Kruskal–Wallis test followed by the Mann–Whitney *U* test with the Bonferroni correction. BBS, Berg Balance Scale; *n*, number; CST, corticospinal tract; CRP, corticoreticular pathway; CPCT, cortico-ponto-cerebellar tract; FA, normalized fractional anisotropy; FN, normalized fiber number.

**Table 3 brainsci-11-00867-t003:** FN and FA values of CST, CRP, CPCT by group according to the independence of ambulation via FAC scores on 1 month after the stroke onset.

		Nonfunctional (FAC 0,1, *n* = 19)	Dependent (FAC 2,3, *n* = 6)	Independent (FAC 4,5, *n* = 2)	*p*-Value
CST	FN	0.406 (0.195–0.571)	0.335 (0.119–0.559)	0.665 (0.389–0.942)	0.870
	FA	0.862 (0.788–0.947)	0.962 (0.939–0.986)	0.952 (0.950–0.954)	0.173
CRP	FN	0.299 (0.204–0.510)	0.346 (0.303–0.408)	0.348 (0.206–0.491)	0.877
	FA	0.977 (0.933–0.988)	0.958 (0.948–0.971)	0.924 (0.919–0.929)	0.330
CPCT	FN	0.118 (0.0–0.442)	0.0 (0.0–0.259)	0.289 (0.145–0.434)	0.692
	FA	0.952 (0.0–1.003)	0.0 (0.0–0.713)	0.458 (0.229–0.686)	0.331

Values are the median (interquartile range: first–third quartiles), FN and FA are calculated via (affected value/non-affected value). *p*-values were tested using Kruskal-Wallis test followed by the Mann-Whitney U-test with the Bonferroni correction. FAC, Functional Ambulation Categories; *n*, number; CST, corticospinal tract; CRP, corticoreticular pathway; CPCT, corticopontocerebellar tract; FA, normalized fractional anisotropy; FN, normalized fiber number.

**Table 4 brainsci-11-00867-t004:** FN and FA values of CST, CRP, CPCT by group according to the independence of ambulation via FAC scores on 6 months after the stroke onset.

		Nonfunctional (FAC 0,1, *n* = 9)	Dependent (FAC 2,3, *n* = 10)	Independent (FAC 4,5, *n* = 8)	*p*-Value
CST	FN	0.159 (0.014–0.883)	0.410 (0.290–0.513)	0.403 (0.121–0.546)	0.859
	FA	0.867 (0.535–0.947)	0.932 (0.836–0.967)	0.905 (0.848–0.953)	0.634
CRP	FN	0.299 (0.269–0.485)	0.344 (0.216–0.451)	0.330 (0.174–0.643)	0.998
	FA	0.976 (0.922–1.008)	0.971 (0.950–0.981)	0.954 (0.929–0.978)	0.703
CPCT	FN	0.073 (0.0–0.533)	0.0 (0.0–0.115)	0.348 (0.121–0.615)	0.163
	FA	0.918 (0.0–0.999)	0.0 (0.0–0.962)	0.957 (0.686–1.015)	0.405

Values are the median (interquartile range: first–third quartiles), FN and FA are calculated via (affected value/non-affected value). *p*-values were tested using Kruskal-Wallis test followed by the Mann-Whitney U-test with the Bonferroni correction. FAC, Functional Ambulation Categories; *n*, number; CST, corticospinal tract; CRP, corticoreticular pathway; CPCT, corticopontocerebellar tract; FA, normalized fractional anisotropy; FN, normalized fiber number.

**Table 5 brainsci-11-00867-t005:** Correlation of DTI parameters and functional assessment scores on 1 month and 6 months after the stroke onset.

	CST		CRP		CPCT	
	FN	FA	FN	FA	FN	FA
BBS on 6 month	0.106	0.329	−0.049	−0.138	−0.016	−0.078
	0.497	0.035	0.751	0.375	0.924	0.634
FAC on 1 month	0.004	0.285	0.033	−0.214	−0.116	−0.245
	0.979	0.074	0.834	0.181	0.492	0.146
FAC on 6 month	0.027	0.121	−0.014	−0.104	0.152	0.091
	0.859	0.436	0.929	0.504	0.351	0.576

FN and FA are calculated via (affected value/non-affected value). T_b_-values and *p*-values were tested using Kendall’s tau b correlation. DTI, Diffusion Tensor Imaging; BBS, Berg Balance Scale; FAC, Functional Ambulation Categories; CST, corticospinal tract; CRP, corticoreticular pathway; CPCT, corticopontocerebellar tract; FA, normalized fractional anisotropy; FN, normalized fiber number.

## Data Availability

The data presented in this study are available on request from the corresponding author.

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
