# Peer review of "Does Motor Tract Integrity at 1 Month Predict Gait and Balance Outcomes at 6 Months in Stroke Patients?"

_brainsci, 2021, doi:10.3390/brainsci11070867_

Round 1

Reviewer 1 Report

This was a very clean, well-designed manuscript evaluating the prediction of gait and balance outcomes at 6 months post-stroke based on motor tract integrity at one-month post-stroke.

Major Concerns

  • given that the title states prediction, I was expecting some type of regression analysis as the current analysis does not relate to prediction. Simple bivariate correlations would have been helpful to know the relationship between variables, and regressions would have been able to answer the primary question. If the authors choose to not re-run the statistics, then the title needs to be changed and a rationale for the current statistics needs to be clearly stated.
  • What is the value in dividing the outcome measures at the given thresholds? Is it simply to divide into thirds, or is there some value for the given metrics? For example, there are thresholds dichotomizing the BBT into risk for falls or not, but I am unaware of clinical value at the stated cut-off points.

Reviewer 2 Report

Summary

Jun et al. investigated the relation between white matter tract integrity and recovery of gait and balance after stroke in 27 patients. They correlated DTI measures of the corticospinal tract (CST), the cortico-reticular pathway (CRP), and the cortico-ponto-cerebellar tract (CPCT) at 1 month post-stroke with gait scores extracted from the Functional Ambulation Categories (FAC) and the Berg Balance Scale (BBS) at 6 months post-stroke. They found that the degree of integrity of these white matter tracts, quantified by fiber number (FN) and fractional anisotropy (FA), did not predict behavioral outcome at 6 months.

Major points:

I think three major points (A, B, C) need clarification: the first two points concern the Methods, the third the Discussion.

A) Prior to concluding that there was a negative result (absence of evidence for a lesion-symptom relation) the authors should, according to me, perform and report several (statistical) tests, to ascertain that their data allow for a valid lesion-symptom assessment. They are spelled out below (points A1 to A5).

B) The authors do not provide any information on the robustness of their DTI measures (FN and FA). It remains therefore unclear whether their negative results hold for this particular definition of DTI seed and target areas (for each tract), or whether slight changes in defining these white matter tracts would change their finding. Without any measure of robustness the validity of the result remains questionable (see B1 below).

C) The Discussion does not adequately relate their findings to those obtained in previous comparative studies. The authors claim that their results differ only little with respect to previous studies. I do not agree with this reading of the literature (see C1, C2 below).

A) Methods:

There are several control measures and prerequisites to establish prior to any testing for a relation between stroke lesion and degree of recovery (6 month status) within the sample:

A1) control 1: was there a significant difference between FN of the affected/unaffected side? This would help establish the validity of this DTI measure.

A2) control 2: was there a significant difference between FA of the affected/unaffected side? This would help establish the validity of this DTI measure.

A3) control 3: was the BBS score independent from the FAC score (absence of correlation)?

A4) prerequisite 1: was there significant recovery in their sample? The authors do not provide statistical evidence for improved BBS and FAC scores over time. Comparing Table 3 and 4 only provides non-statistical evidence for improved FAC score.

A5) prerequisite 2: was there a significant relation between lesion (CST, CRP, and CPCT integrity) and symptom severity at 1 month? What would the rationale be for a relation to recovery (6 month status) if this prerequisite cannot be ascertained? The authors only provide these data for FAC (Table 3), but not for BBS. And there was no relation of FA or FN to FAC score.

Points A1) to A5) need, according to me, a positive statistical answer prior to investigating the lesion-symptom (recovery) relation at 6 months.

B1) The validity and robustness of FA and FN (for CST, CRP, and CPCT) as a function of seed and target area definition (regions of interest, ROIs) should be demonstrated. In other words, how reliable (variable) are FN and FA data when changing, within reasonable measures, the  definition of the respective seed and target ROIs? Since there are no hard criteria for defining these ROIs this seems a legitimate and critical (inter-rater) issue. Although Fig1-3 indicate the chosen rostro-caudal height of the seed ROIs, this choice is arbitrary within certain limits. And the extent (medio-lateral and antero-posterior area, Fig. 1B,C; Fig. 2B,C,D; Fig. 3B,C) of the ROI may also affect FA and FN measures. In addition, how the ROIs were defined is not explained in Methods: this needs improvement. Were they drawn on the non-lesioned side and then flipped to the other side? And for the CST : why were pontine ROIs chosen and not in addition the internal capsule?

C1) Discussion: the current discussion only mentions the small number of subjects as a potential reason for their negative finding. This is insufficient and inadequate. Other variables come into play, such as time of follow-up, type of behavioral characterization (different clinical scales with limits in sensitivity, potential floor and ceiling effects), methodological differences in DTI parameters (seed, target areas and others). These points should be mentioned and put into context.   

C2) Discussion: the authors claim that there are only “small differences in results between previous studies and our investigation » (line 192-193). I do not agree, in particular with respect to two studies that found clear relations between post-stroke CST integrity and gait:

Jones et al., 2016 (not cited) showed that CST damage independently predicted response to therapy for FAC and MRMI (at 6 weeks follow-up). Walking speed was predicted by damage to the putamen, insula, external capsule and neighbouring white matter (but not CST).

Soulard et al., 2020 (cited, ref 5) found clear relations between 2 year follow-up walking speed and CST integrity and corticoreticular pathway integrity.

I consider the present results contradictory to those of Jones et al., 2016 and of Soulard et al., 2020. This needs thorough discussion.

Minor points:

They authors may want to (should) cite: Jones PS, Pomeroy VM, Wang J, Schlaug G, Tulasi Marrapu S, Geva S, Rowe PJ, Chandler E, Kerr A, Baron JC; SWIFT-Cast investigators. Does stroke location predict walk speed response to gait rehabilitation? Hum Brain Mapp. 2016 Feb;37(2):689-703. doi: 10.1002/hbm.23059. (see point C2 above).

Fig. 1A,B,C. Indicating ‘a’ (anterior) and ‘p’ (posterior) would help.

Fig. 2B,C and D are presumably (Methods line 93 ) B0 images. This should be spelled out in the legend.

Fig. 2D. Is the image distorted? The medio-lateral dimension seems large compared to the antero-posterior. I cannot see the M1 hand knob, with respect to which I would position area 6. Without this landmark it is difficult for the reader to appreciate correct positioning and shape/size of the BA6 ROI target area.

Fig. 2. The legend says line 134 and 136 “… in the axial color map …”. This seems to be wrong: the color map in A is sagittal. The orientation (axial, coronal, sagittal) should be spelled out for each image (here and in legend to Fig1 and 3).

Fig. 2. Please mention Jang and Lee 2019 (ref 9) in the legend when defining the CRP seed and target ROIs (in particular the use of area 6).

Fig. 1D, 2E and Fig3D: for more clarity, the authors may consider superposing the DTI tracts with B0 images rather than presenting the respective DTT image.

Round 2

Reviewer 1 Report

The authors have adequately addressed the concerns that I raised in my original review. In addition, I read the other excellent review and the responses, and I have no additional concerns at this time.

Reviewer 2 Report

Thanks for constructive answers to most major and minor comments and for respective revisions in the Ms. Suggest some minor editing (below).

Minor editing/wording:

line 133: suggest replacing “… CRP and CPCT. (All p value was less than 0.001)” by “… CRP and CPCT (all p-values < 0.001).”

line 160 (legend to Fig. 2): delete “Jang and Lee [9] was cited for defining of ROIs” and simply add “[9]“ at the end of the first sentence (line 153) such that “… of the CRP [9]“.

line 199 (Title to Table 5): should probably read “Correlation of DTI …” and not “… DTIT … ”.

line 202 (Legend to Table 5): “n, number;” can probably be deleted (what does it refer to?).  

line 219: suggest changing “These findings were differing from some previous studies”  to “These findings differ from those of some previous studies”. 

line 240-241 (Conclusion): suggest alternative wording “The integrity of the CST, CRP, and CPCT, as assessed by DTI about 1 month after a supratentorial first-ever stroke, did not reflect the functional outcome of gait and balance after 6 months.”

Author Response

Dear reviewer

The authors thank the editor and reviewers for their careful evaluation of our manuscript. We believe that our manuscript “Does motor tract integrity at 1 month predict gait and balance outcomes at 6 months in stroke patients?” has been improved by the insightful comments and suggestions. We have carefully considered the suggestions and have addressed all comments, point-by-point in the revised manuscript. The changes are summarized below.

Reviewer #2:

line 133: suggest replacing “… CRP and CPCT. (All p value was less than 0.001)” by “… CRP and CPCT (all p-values < 0.001).”

Answer) We have revised that as recommended.

line 160 (legend to Fig. 2): delete “Jang and Lee [9] was cited for defining of ROIs” and simply add “[9]“ at the end of the first sentence (line 153) such that “… of the CRP [9]“.

Answer) We have modified the figure legends as recommended.

line 199 (Title to Table 5): should probably read “Correlation of DTI …” and not “… DTIT … ”.

Answer) We have changed it as recommended.

line 202 (Legend to Table 5): “n, number;” can probably be deleted (what does it refer to?). 

Answer) We have done as the reviewer suggested.

line 219: suggest changing “These findings were differing from some previous studies”  to “These findings differ from those of some previous studies”.

Answer) We have converted that sentence as recommended.

line 240-241 (Conclusion): suggest alternative wording “The integrity of the CST, CRP, and CPCT, as assessed by DTI about 1 month after a supratentorial first-ever stroke, did not reflect the functional outcome of gait and balance after 6 months.”

Answer) We have done that in the revised manuscript as suggested.

We hope that the changes are satisfactory and the manuscript is now suitable for publication in Brain Sciences. We are very grateful for your time and efforts in dealing with our manuscript.

Sincerely yours,

Jun 24, 2021

Seong Hoon Lim, M.D., Ph.D.

Department of Rehabilitation Medicine, St. Vincent’s Hospital

College of Medicine, The Catholic University of Korea

Ji-dong 93, Paldal-gu, Suwon, 442-723 Republic of Korea

Telephone number- +82-31-249-7650

Email- seonghoon@catholic.ac.kr/limseonghoon@gmail.com
